# Severity of COVID-19 at elevated exposure to perfluorinated alkylates

Philippe Grandjean[1,2]*, Clara Amalie Gade Timmermann[2], Marie Kruse[2], Flemming Nielsen[2], Pernille Just Vinholt[3], Lasse Boding[4], Carsten Heilmann[5], Kåre Mølbak[4,6]

1 The Department of Environmental Health, Harvard T.H. Chan School of Public Health, Boston, Massachusetts, United States of America, 2 The Department of Public Health, University of Southern Denmark, Odense, Denmark, 3 The Department of Clinical Biochemistry and Pharmacology, Odense University Hospital, Odense, Denmark, 4 The Department of Epidemiology, Statens Serum Institut, Copenhagen, Denmark, 5 Pediatric Clinic, Rigshospitalet - National University Hospital, Copenhagen, Denmark, 6 The Department of Veterinary and Animal Sciences, Faculty of Health and Medical Sciences, University of Copenhagen, Copenhagen, Denmark

* pgrand@hsph.harvard.edu

**Data Availability Statement:** Data cannot be shared publicly because of the need to protect personal data. Anonymous data are available from

## Abstract

### Background

The course of coronavirus disease 2019 (COVID-19) seems to be aggravated by air pollution, and some industrial chemicals, such as the perfluorinated alkylate substances (PFASs), are immunotoxic and may contribute to an association with disease severity.

### Methods

From Danish biobanks, we obtained plasma samples from 323 subjects aged 30–70 years with known SARS-CoV-2 infection. The PFAS concentrations measured at the background exposures included five PFASs known to be immunotoxic. Register data was obtained to classify disease status, other health information, and demographic variables. We used ordered logistic regression analyses to determine associations between PFAS concentrations and disease outcome.

### Results

Plasma-PFAS concentrations were higher in males, in subjects with Western European background, and tended to increase with age, but were not associated with the presence of chronic disease. Of the study population, 108 (33%) had not been hospitalized, and of those hospitalized, 53 (16%) had been in intensive care or were deceased. Among the five PFASs considered, perfluorobutanoic acid (PFBA) showed an unadjusted odds ratio (OR) of 2.19 (95% confidence interval, CI, 1.39–3.46) for increasing severities of the disease. Among those hospitalized, the fully adjusted OR for getting into intensive care or expiring was 5.18 (1.29, 20.72) when based on plasma samples obtained at the time of diagnosis or up to one week before.

the secure server at the Danish Health Data
Authority, pending necessary approvals from the
Authority (instructions at www.sundhedsdata.dk)
and the Regional Committee on Health Research
Ethics for researchers who meet the criteria for
access to confidential data. Aggregated data
underlying the results presented in the study are
available from the corresponding author, provided
that no data are based on less than 5 individuals.

**Funding:** Novo Nordisk Foundation
(NNF20SA0062871) (LB, KM). National Institute
for Environmental Health Sciences (ES027706)
(PG). The funders had no role in study design, data
collection and analysis, decision to publish, or
preparation of the manuscript.

**Competing interests:** Apart from PG having served
as health expert in lawsuits on environmental
contamination, which does not affect the
adherence to all PLOS ONE policies, the authors
have no competing interests to declare, financial or
otherwise.

## Conclusions

Measures of individual exposures to immunotoxic PFASs included short-chain PFBA known
to accumulate in the lungs. Elevated plasma-PFBA concentrations were associated with an
increased risk of a more severe course of COVID-19. Given the low background exposure
levels in this study, the role of exposure to PFASs in COVID-19 needs to be ascertained in
populations with elevated exposures.

## Introduction

Elevated exposure to community pollution is associated with a worsened outcome of coronavi-
rus disease 2019 (COVID-19) [1–4]. While replicated in different populations, this evidence
relies solely on ecological study designs of air pollution without measures of individual expo-
sures. Several environmental chemicals are known to suppress immune functions [5, 6] and
worsen the course of infections [7]. Of particular relevance, the perfluorinated alkylate sub-
stances (PFASs) are persistent, globally disseminated chemicals known to be immunotoxic [8].
Thus, elevated blood-PFAS concentrations are associated with lower antibody responses to
vaccinations in children [9] and in adults [10]. Also, infectious disease occurs more frequently
in children with elevated exposure [11–13]. In support of the potential impact of these sub-
stances, a modeling study suggested that endocrine disruptors, including major PFASs, may
interfere with proteins involved in critical pathways, such as IL-17, associated with severe clini-
cal outcomes of the COVID-19 infection [14].

Substantial differences occur in the clinical course of the disease, and the reasons for this
variability are only partially known [15, 16]. As a possible contributor, a deficient antibody
response may be an important contributor to a more severe clinical course of the infection
[17], as also suggested by the poorer prognosis in patients with bacterial co-infection [18]. The
most serious clinical consequences are associated with male sex, older age, and the presence of
co-morbidities, including obesity and diabetes [19–23]. In parallel, serum-PFAS concentra-
tions are higher in men than in women and also tend to increase with age [8, 24]. Because ele-
vated PFAS exposure has been linked to both obesity and diabetes [25, 26], these substances
may potentially affect the progression of COVID-19 directly as well as indirectly.

Several PFASs can be reliably determined in human blood samples, where most of them
show long biological half-lives of 2–3 years or more [27], thereby providing a measure of
cumulated exposure. Still, blood concentrations may not accurately reflect the retention in spe-
cific organs, e.g., the short-chain perfluorobutanoic acid (PFBA), which accumulates in the
lungs [28].

To assess if elevated background exposures to immunotoxic PFASs are associated with the
clinical course of the infection, a study was undertaken in Denmark to determine individual
plasma-PFAS concentrations in adults confirmed to be infected with severe acute respiratory
syndrome coronavirus 2 (SARS-CoV-2) and examine the association with the severity of
COVID-19 development.

## Methods

### Population

Plasma samples for PFAS analysis were obtained from medical biobanks that store excess
material from diagnostic tests, viz., the Danish National Biobank at the Statens Serum Institut

(SSI) and Odense University Hospital (OUH). Eligible subjects were identified from the Danish cohort of COVID-19 patients [29]. All cases were tested by quantitative polymerase-chain-reaction (PCR) and had a positive response for SARS-CoV-2 infection, as recorded in the Danish Microbiology Database (MiBa), a national database that contains both positive and negative results of the majority of microbiology testing done in Denmark [30].

The study included non-pregnant subjects aged 30–70 years at the time of the positive test by early March 2020 through early May 2020, provided that the biobanks could provide a plasma sample of 0.15 mL. Although most blood samples were obtained soon after SARS-CoV-2 infection was identified, we also included subjects, mainly those not hospitalized, whose plasma in the SSI biobank had been obtained up to 28 months earlier, i.e., less than a half-life for major PFASs [27]. We calculated the time interval from blood sampling to the time of diagnosis, of relevance mainly for non-hospitalized subjects. In those hospitalized, we computed the interval from admission to the time of sampling the plasma used for PFAS analysis.

All samples were coded, and the Personal Identification Number for each subject was separately transferred to the Danish Health Data Authority (FSEID-00005000) to allow linkage to demographic and medical information from the Danish Civil Registration System (CRS) [31], the Danish National Register of Patients (DNRP) [32], and the National Health Insurance Service Register [33]. We used the following classification of disease status: no hospital admission and completed infection within 14 days of testing positive, hospitalization with COVID-19 up to, or above, 14 days, admission to intensive care unit, or death. Presence of chronic disease was based on the following diagnoses in the register data: diabetes type I and II (ICD10 codes E10-E11), malignant cancers (C00-C99), cerebrovascular and coronary disease (I00-I99), pulmonary disease (J00-J99), and obesity (E66-E68). Renal disease (N0-N2) was treated as a separate covariate due to the possible impact of kidney function on plasma-PFAS concentrations [34]. The linked data set was analyzed via secure server without access to information on the Personal Identification Numbers of the subjects involved. For confidentiality reasons, all tabular information had to be based on at least five subjects.

The protocol was approved by the Regional Committee on Health Research Ethics (S-20200064), which also allowed the project to proceed without seeking informed consent from the subjects identified for study participation. Additional approvals were obtained from the Danish Data Protection Agency as well as institutional and regional authorities for the transfer blood samples and linkage of subject information to the PFAS analyses, while protecting confidentiality.

## Chemical analysis

The plasma samples were analyzed in successive series for PFAS concentrations, including PFBA, perfluorooctane sulfonate (PFOS), perfluorooctanoate (PFOA), perfluorohexane sulfonate (PFHxS), and perfluorononanoate (PFNA), which are known from previous studies to be associated with immunotoxicity in humans [8, 35, 36]. We also determined plasma concentrations of PFASs so far not linked to immunotoxicity, i.e., short-chain perfluorobutanesulfonate (PFBS), perfluoroheptanesulfonate (PFHpS), perfluorodecanoate (PFDA), and perfluoroundecanoate (PFUdA) (results shown in the Supporting information). We used online solid-phase extraction followed by liquid chromatography and triple quadropole mass spectrometry (LC–MS/MS) at the University of Southern Denmark [37]. Accuracy of the analysis was ensured by inclusion of quality control (QC) samples comprising proficiency test specimens from the HBM4EU program organized by Interlaboratory Comparison Investigations (ICI) and External Quality Assurance Schemes (EQUAS). All results of the QC samples were within the

acceptance range. The between-batch CVs for the actual series ranged between 3% and 14% for all compounds. Both PFOS and PFOA were quantified in all blood samples, and all PFASs were detectable in at least 30% of the samples. Results below the limit of detection (LOD, 0.03 ng/ml) were replaced by LOD/2 before uploading to the secure server at the Danish Health Data Authority, where linkage to other information took place.

## Statistical analysis

Correlations between PFASs were examined using Spearman's correlation coefficient. The PFAS concentrations were compared between demographic groups (age in years, sex, national origin, place of inclusion), presence of comorbidities, and number of days between blood sampling and diagnosis, and differences were tested using Kruskal-Wallis and Wilcoxon rank-sum test. Furthermore, associations of COVID-19 severity with age were tested using Kruskal-Wallis test, and relations with each of the variables sex, national origin, presence of comorbidities, and number of days between blood sampling and diagnosis were tested using $\chi^2$ test. Associations between place of inclusion and COVID-19 severity could not be displayed and tested, as some cells contained less than five individuals.

Because COVID-19 severity was categorized, the association between the continuous plasma-PFAS concentrations and COVID-19 severity was tested in ordered logistic regression models. More than half the short-chain PFAS concentrations were below the LOD, and they were therefore treated as binary variables (below/above LOD). Potential confounding variables were identified based on a priori knowledge as summarized above and included age (continuous, years) sex, and national origin (Western European yes/no). Among those of Western European national origin, 94% were Danish, while most of the participants of non-Western European national origin were born in or of parents from Somalia (20% of the sample), Pakistan (13%), Iraq (12%), Morocco (11%), Eastern Europe (9%), and Turkey (9%). Kidney disease may affect PFAS elimination, and PFAS exposure could potentially increase the risk of certain other chronic diseases that may affect COVID-19 severity [8]. Kidney disease (yes/no) and other chronic disease (yes/no) were thus considered potential confounders to allow estimation of the direct, rather than the total effect of plasma-PFAS concentrations. Due to changes in PFAS exposures over time, the timing of blood sampling was included as covariate. Further, due to the short elimination half-life for short-chain PFASs [8], we carried out sensitivity analyses excluding plasma samples obtained more than one week before or after diagnosis. We also adjusted for the place of inclusion (OUH/SSI) but, under the circumstances of this study, detailed data on socioeconomic status (e.g., income, education or labor market affiliation) were unavailable for this study. Dichotomous analyses comparing severities of the disease were performed in logistic regression models.

The default assumption of dose-response linearity was tested by including PFAS squared along with PFAS in the regression models. No significant (p<0.05) deviation from linearity was found. The proportional odds assumption in the ordered logistic regression was tested by a likelihood-ratio test using the Stata *omodel* package. In a model adjusting for age, place of inclusion, and timing of blood sampling, the hypothesis of proportional odds was accepted (p>0.05) in all analyses. Odds ratios (ORs) between groups of COVID-19 severity were therefore calculated using logistic regression models.

## Results

The predominant PFAS in plasma was PFOS, with an average concentration of 6.1 ng/mL (median, 4.7 ng/L), approximately equally distributed between the normal and branched isomers. Other PFASs quantified showed averages below 1 ng/mL. In a sensitivity analysis, one

**Table 1. Spearman's correlation coefficients for pairwise comparisons of detectable PFASs in plasma from 323 subjects included in the study.**

|        | PFBA   | PFHxS  | PFOA   | PFOS   |
|--------|--------|--------|--------|--------|
| PFHxS  | 0.0520 |        |        |        |
| PFOA   | 0.0617 | 0.7072 |        |        |
| PFOS   | 0.0591 | 0.8406 | 0.7248 |        |
| PFNA   | 0.0127 | 0.7133 | 0.7759 | 0.8406 |

extreme PFHxS outlier at 12.9 ng/mL was omitted. The PFAS concentrations correlated well, with Spearman correlation coefficients generally above 0.5 (Table 1 and S1 Table), except for short-chained PFAS. PFOS on average contributed 69% of the total PFAS concentrations by weight and correlated particularly well with most other PFASs quantified.

In general, serum-PFAS concentrations were higher at older ages, in men, and among those of Western European origin. Although the presence of chronic disease did not seem to be associated with PFAS, the plasma concentrations appeared to be higher in the presence of kidney disease (Table 2 and S2 Table).

In the study population, males, older subjects, and those with chronic disease, were more frequently represented among subjects with severe COVID-19, while there was no difference in regard to national origin for disease severity (Table 3). The PFAS-associations with disease severity were similar in Western Europeans and subjects with other backgrounds ($P > 0.2$ for population differences).

A more severe disease outcome was associated with higher plasma-PFBA concentrations, also after adjustment for all covariates (Table 4 and S4 Table). None of the other PFASs showed a similar tendency. If leaving out presence of chronic disease as a non-significant predictor, the adjusted OR for PFBA was 1.77 (95% CI, 1.09, 2.87). More importantly, when excluding samples collected earlier than one week before the time of diagnosis (148 samples), or more than one week later (5 samples), stronger ORs emerged for PFBA (Table 4). Counter to the *a priori* hypothesis, some PFASs, including PFHxS, seemed associated with a lower risk, but this tendency was weakened when relying on plasma samples collected in close connection to the diagnosis of corona infection (Table 4 and S3 Table).

In dichotomous analyses comparing severities of the disease (S4 Table), detectable PFBA in plasma also showed a clear association with a more severe clinical course of the disease, most pronounced for odds between hospitalization and admission to intensive care unit/death, especially when based on plasma samples obtained at the time of diagnosis or up to one week before where the adjusted OR was 5.18 (1.29, 20.72). No such tendency was seen for the other PFASs detected (S4 Table). The association between PFBA and disease severity was similar for men and women (Fig 1).

## Discussion

The present study aimed at determining the potential aggravation of COVID-19 associated with elevated exposures to PFASs. Several of these substances are known immunotoxicants in laboratory animals [35] and in humans [8, 9]. In addition to immunotoxicity, major PFASs can potentially interfere with major pathways that are predictive of a serious clinical outcome of the infection [14]. An association of PFAS exposure with disease severity therefore appears biologically plausible.

Among the PFASs, presence of detectable PFBA in plasma showed the strongest positive association with the severity of the disease. This finding may at first seem surprising, as this

**Table 2. Median plasma-PFAS concentrations (25[th], 75[th] percentiles) in ng/mL by population characteristics.**

| Population characteristics | n (%) | PFBA | PFAS (ng/mL) median (25th,75th percentile) | | | |
| --- | --- | --- | --- | --- | --- | --- |
| | | | PFHXS | PFOA | PFOS | PFNA |
| Total | 323 (100) | <LOD (<LOD, 0.04) | 0.48 (0.28, 0.71) | 0.77 (0.43, 1.18) | 4.86 (2.85, 8.29) | 0.38 (0.23, 0.59) |
| **Age (years)** | | | | | | |
| 30–39 | 37 (11) | <LOD (<LOD, 0.03) | 0.32 (0.19, 0.46) | 0.59 (0.43, 0.86) | 3.30 (1.89, 5.27) | 0.29 (0.21, 0.43) |
| 40–49 | 64 (20) | <LOD (<LOD, 0.03) | 0.35 (0.15, 0.57) | 0.58 (0.35, 0.89) | 3.11 (2.24, 5.06) | 0.27 (0.19, 0.39) |
| 50–59 | 106 (33) | <LOD (<LOD, <LOD) | 0.50 (0.31, 0.75) | 0.83 (0.43, 1.18) | 5.41 (2.79, 8.84) | 0.40 (0.24, 0.61) |
| 60–70 | 116 (36) | <LOD (<LOD, 0.05) | 0.56 (0.39, 0.89) | 0.97 (0.56, 1.51) | 6.11 (3.83, 9.60) | 0.48 (0.30, 0.70) |
| p-value [a] | | 0.008 | <0.001 | <0.001 | <0.001 | <0.001 |
| **Sex** | | | | | | |
| Male | 174 (54) | <LOD (<LOD, 0.04) | 0.59 (0.40, 0.87) | 0.81 (0.51, 1.26) | 5.96 (3.65, 10.17) | 0.40 (0.25, 0.61) |
| Female | 149 (46) | <LOD (<LOD, 0.04) | 0.35 (0.17, 0.52) | 0.70 (0.40, 1.04) | 3.43 (2.06, 5.66) | 0.36 (0.22, 0.56) |
| p-value [b] | | 0.713 | <0.001 | 0.011 | <0.001 | 0.131 |
| **Kidney disease** | | | | | | |
| yes | 34 (11) | <LOD (<LOD, 0.06) | 0.55 (0.34, 0.77) | 0.91 (0.54, 1.46) | 5.60 (3.08, 8.38) | 0.50 (0.24, 0.67) |
| no | 289 (89) | <LOD (<LOD, 0.03) | 0.47 (0.28, 0.71) | 0.76 (0.43, 1.15) | 4.76 (2.82, 8.10) | 0.36 (0.23, 0.57) |
| p-value [b] | | 0.040 | 0.466 | 0.065 | 0.489 | 0.141 |
| **Other chronic disease** | | | | | | |
| Yes | 220(68) | <LOD (<LOD, 0.04) | 0.47 (0.28, 0.68) | 0.71 (0.42, 1.15) | 4.70 (2.87, 7.99) | 0.38 (0.23, 0.57) |
| No | 103 (32) | <LOD (<LOD, 0.03) | 0.51 (0.28, 0.76) | 0.87 (0.47, 1.23) | 5.35 (2.72, 8.41) | 0.41 (0.23, 0.65) |
| p-value [b] | | 0.075 | 0.314 | 0.124 | 0.850 | 0.407 |
| **National origin** | | | | | | |
| Western Europe | 224 (69) | <LOD (<LOD, 0.04) | 0.52 (0.35, 0.76) | 0.91 (0.60, 1.29) | 5.61 (3.40, 9.18) | 0.43 (0.29, 0.64) |
| Other | 99 (31) | <LOD (<LOD, 0.04) | 0.34 (0.16, 0.57) | 0.44 (0.31, 0.80) | 2.86 (1.61, 5.13) | 0.23 (0.16, 0.36) |
| p-value [b] | | 0.552 | <0.001 | <0.001 | <0.001 | <0.001 |
| **Place of inclusion** | | | | | | |
| Odense | 48 (15) | <LOD (<LOD, 0.06) | 0.45 (0.32, 0.69) | 0.67 (0.42, 0.95) | 4.67 (3.29, 8.09) | 0.36 (0.24, 0.45) |
| Copenhagen | 275 (85) | <LOD (<LOD, 0.03) | 0.48 (0.28, 0.72) | 0.79 (0.44, 1.20) | 4.89 (2.72, 8.31) | 0.39 (0.23, 0.62) |
| p-value [b] | | 0.003 | 0.967 | 0.203 | 0.697 | 0.299 |
| **Timing of blood sampling** | | | | | | |
| After diagnosis—1 week before | 193 (60) | <LOD (<LOD, 0.04) | 0.48 (0.30, 0.71) | 0.70 (0.40, 1.11) | 4.63 (2.83, 7.65) | 0.34 (0.23, 0.56) |
| >1 week—1 year before | 46 (14) | <LOD (<LOD, 0.03) | 0.45 (0.21, 0.66) | 0.82 (0.38, 1.35) | 4.81 (2.36, 8.62) | 0.38 (0.20, 0.65) |
| > 1year before diagnosis | 84 (26) | <LOD (<LOD, 0.03) | 0.50 (0.30, 0.72) | 0.87 (0.57, 1.22) | 5.48 (3.10, 10.28) | 0.45 (0.28, 0.65) |
| p-value [a] | | 0.185 | 0.756 | 0.085 | 0.209 | 0.053 |

[a] Variables with more than two categories tested using Kruskal-Wallis rank test.

[b] Binary variables tested using Wilcoxon rank-sum test.

PFAS has a short elimination half-life in the blood and is often considered of less importance to health [27]. However, in tissue samples from autopsies, PFBA is the only PFAS that is substantially accumulated in the lungs [28]. Given the persistence of the PFASs in general, the unique retention of PFBA in lung tissue may offer a clue to interpreting the findings in this study.

Some odds ratios for PFBA were weakened after adjustment for covariates. However, adjustment for all covariates may result in over-adjustment bias. Thus, older age and male sex are known to be strong predictors of higher blood-PFAS concentrations, and simple adjustment for these factors could potentially result in a bias toward the null. As PFAS exposure has been linked to important comorbidities, such as diabetes and obesity [25, 26], both of which

**Table 3. COVID-19 severity by population characteristics.**

| | | COVID-19 severity | | |
|---|---|---|---|---|
| Population characteristics | No. of subjects | No hospitalization | Hospitalization | Intensive care unit and/or deceased |
| **Total No. of subjects (%)** | 323 (100) | 108 (33) | 162 (50) | 53 (16) |
| **Age (years)** median (25th,75th percentile) | 55 (46, 62) | 49 (41, 57) | 57 (51, 63) | 62 (53, 67) |
| P value [a] | <0.001 | | | |
| **Sex** | | | | |
| Male, n (%) | 174 (100) | 44 (25) | 94 (54) | 36 (21) |
| Female, n (%) | 149 (100) | 64 (43) | 68 (46) | 17 (11) |
| P value [b] | 0.002 | | | |
| **Kidney disease** | | | | |
| Yes, n (%) | 34 (11) | 7 (21) | 13 (38) | 14 (41) |
| No, n (%) | 289 (89) | 101 (35) | 149 (52) | 39 (13) |
| P value [b] | <0.001 | | | |
| **Other chronic disease** | | | | |
| Yes, n (%) | 220 (100) | 54 (25) | 119 (54) | 47 (21) |
| No, n (%) | 103 (100) | 54 (52) | 43 (42) | 6 (6) |
| P value [b] | <0.001 | | | |
| **National origin** | | | | |
| Western Europe, n (%) | 224 (100) | 76 (34) | 113 (50) | 35 (16) |
| Other, n (%) | 99 (100) | 32 (32) | 49 (49) | 18 (18) |
| P value [b] | 0.844 | | | |
| **Days between blood sampling and diagnosis** | | | | |
| median (25th,75th percentile) | 0 (-1, 393) | 335 (22.5, 639.5) | 0 (-1, 0) | 0 (-2, 1) |
| P value [a] | <0.001 | | | |

[a] Associations tested using Kruskal-Wallis rank test.

[b] Associations tested using Pearson's chi-squared test.

may exacerbate the virus infection, adjustment for chronic disease may also not be justified. Leaving it out slightly strengthened the PFBA association with the disease severity. The strongest associations for PFBA, but not for other PFASs, appeared when focusing on the most representative blood samples obtained close to the time of diagnosis.

**Table 4. Ordered logistic regression OR of increased Covid-19 severity for an increase by 1 ng/mL in plasma-PFAS concentrations.**

| PFAS | No. of subjects | OR (95% CI) | | No. of subjects | OR (95% CI) |
|---|---|---|---|---|---|
| | | Crude | Adjusted for main covariates[a] | | Exposure at time of diagnosis[a,b] |
| PFBA (>LOD/<LOD) | 104/219 | 2.19 (1.39, 3.46) | 1.57 (0.96, 2.58) | 61/109 | 2.10 (1.02, 4.33) |
| PFHxS (ng/mL) | 323 | 0.85 (0.63, 1.15) | 0.52 (0.29, 0.91) | 170 | 0.52 (0.24, 1.14) |
| PFHxS [c] (ng/mL) | 322 | 1.00 (0.62, 1.61) | 0.52 (0.29, 0.93) | 169 | 0.53 (0.22, 1.27) |
| PFOA (ng/mL) | 323 | 0.99 (0.72, 1.36) | 0.83 (0.57, 1.20) | 170 | 0.62 (0.36, 1.08) |
| PFOS (ng/mL) | 323 | 1.00 (0.96, 1.04) | 0.97 (0.92, 1.02) | 170 | 0.98 (0.89, 1.07) |
| PFNA (ng/mL) | 323 | 1.18 (0.67, 2.09) | 1.04 (0.54, 2.02) | 170 | 0.73 (0.25, 2.11) |

[a] Adjusted for age, sex, kidney disease, other chronic disease, national origin, place of testing, and days between blood sampling and diagnosis.

[b] Excluding individuals who had blood sampled more than one week before or after diagnosis.

[c] PFHxS >10 ng/mL excluded.

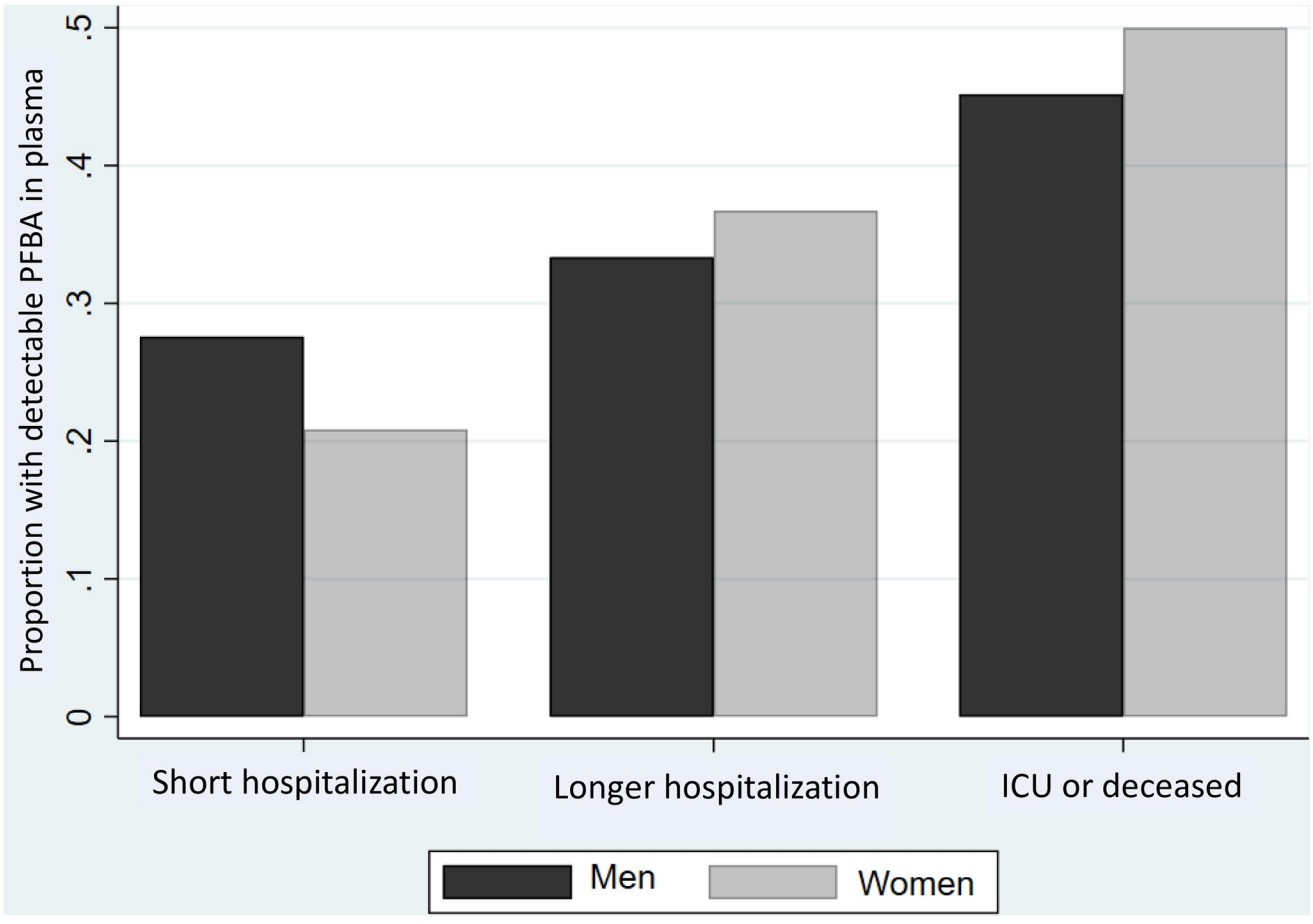

**Fig 1. Proportion of plasma samples with detectable PFBA concentrations at different disease severities.** Results are shown for 44 men and 64 women with up to two weeks of hospitalization, 94 men and 68 women with longer hospitalization, and 36 men and 17 women admitted to the intensive care unit (ICU) or deceased ($P = 0.003$).

An additional consideration is that the present study relates to low background exposure levels, in comparison with PFAS concentrations to findings in, e.g., U.S. adults [38]. Given the wide occurrence of highly contaminated drinking water in other countries [39], the present study results should not be interpreted as evidence that most PFASs do not contribute to a worsened clinical course of COVID-19.

The results for PFBA in this study appear to parallel the findings in regard to other environmental toxicants, viz., air pollutants [1–4] and suggest a need to ascertain the impact of relevant occupational or environmental exposures on COVID-19 severity. Of note, the evidence on air pollution relies solely on ecological study designs without measures of individual levels of exposure, while the present study benefitted from measurements of plasma-PFAS concentrations of all study subjects.

In regard to limitations, the study population may not be representative of corona-positive subjects, as inclusion in the study depended solely on the existence of plasma from diagnostic blood samples at the participating hospitals. Thus, subjects with chronic disease or more severe COVID-19 likely had more frequent hospital visits or longer admissions and thereby a greater chance of having plasma available for inclusion in this study. With a corona-related fatality rate of Danish blood donors below 70 years of age at 89 per 100,000 infections [40], the

presence of 17 deaths in the present material (i.e., against 0.3 deaths expected) confirms that the blood samples represent a highly selected population. Still, a total of 108 subjects were known to have been infected, though not hospitalized. In many cases, their plasma had been stored on previous occasions, and the PFAS concentrations may reflect slightly higher exposures in the recent past [8], which could possibly explain the apparent protective effects of some PFASs, although adjustment for the time interval since sample collection was included in the analyses. However, the strongest associations for PFBA, but not for other PFASs, were seen when excluding samples not obtained in close temporal connection with the infection.

The study population included mostly older subjects who were more frequently male, and a large proportion of foreign-born subjects and second-generation immigrants (Table 3), thereby possibly deviating from the background population of corona-infected patients in Denmark. Still, the results do not suggest major biases affecting PFAS exposure and its association with COVID-19 outcomes.

Among immigrants, adverse associations appeared slightly stronger, also after adjustments, in accordance with national origin, perhaps as related to demographic or social factors, resulting in a greater likelihood also to PFAS-associated aggravation of the infection. Difference in age, sex, or comorbidities did not explain this tendency, but is in agreement with previous findings of ethnic differences in vulnerability [41]. However, national origin may be a surrogate marker for other factors, such as exposure at work or exposure within crowded households, as immigrant origin tends to be associated with certain occupations including front-line workers and living in areas with higher population density [42]. Still, in agreement with higher PFAS exposure being associated with higher socioeconomic position [43], we found that the association between PFBA exposure and disease severity was independent of national origin.

## Conclusions

Increased plasma-PFBA concentrations were associated with a greater severity of COVID-19 prognosis, and this tendency remained after adjustment for sex, age, comorbidities, national origin, sampling location and time. Although occurring in fairly low concentrations in plasma, PFBA is known to accumulate in the lungs. Thus, as immunotoxic substances, the PFASs may well contribute to the severity of COVID-19. The present findings on a short-chain PFAS at background exposures suggest a need to ascertain if elevated exposures to environmental immunotoxicants may worsen the outcome of the SARS-CoV-2 infection.

## Supporting information

**S1 Table. Spearman's correlation coefficients for pairwise comparisons of detectable PFASs in plasma from 323 subjects included in the study.**
(DOCX)

**S2 Table. Median plasma concentrations of additional PFASs (25th-75th percentiles) in ng/mL by population characteristics.**
(DOCX)

**S3 Table. Ordered logistic regression odds ratios (ORs) of increased COVID-19 severity at an increase by 1 ng/mL in plasma concentrations of additional PFASs.**
(DOCX)

**S4 Table. Logistic regression odds ratios (ORs) of increased COVID-19 severity at an increase by 1 ng/mL in plasma concentrations of all PFASs detected.**
(DOCX)

## Acknowledgments

The Danish Departments of Clinical Microbiology, the Danish Microbiology Database and the section for Data Integration and Analysis at Statens Serum Institut collated the national COVID-19 data. Plasma samples were identified and provided by Statens Serum Institut and the Department of Clinical Biochemistry and Pharmacology at Odense University Hospital.

## Author Contributions

**Conceptualization:** Philippe Grandjean, Carsten Heilmann, Kåre Mølbak.

**Data curation:** Philippe Grandjean, Clara Amalie Gade Timmermann, Marie Kruse, Flemming Nielsen.

**Formal analysis:** Philippe Grandjean, Clara Amalie Gade Timmermann, Marie Kruse.

**Funding acquisition:** Philippe Grandjean, Kåre Mølbak.

**Investigation:** Philippe Grandjean, Clara Amalie Gade Timmermann, Marie Kruse, Flemming Nielsen, Pernille Just Vinholt, Lasse Boding.

**Methodology:** Philippe Grandjean, Clara Amalie Gade Timmermann, Marie Kruse, Flemming Nielsen, Pernille Just Vinholt, Lasse Boding.

**Project administration:** Philippe Grandjean, Kåre Mølbak.

**Resources:** Philippe Grandjean, Lasse Boding, Kåre Mølbak.

**Software:** Clara Amalie Gade Timmermann, Marie Kruse.

**Supervision:** Philippe Grandjean, Kåre Mølbak.

**Validation:** Philippe Grandjean, Clara Amalie Gade Timmermann.

**Visualization:** Marie Kruse.

**Writing – original draft:** Philippe Grandjean.

**Writing – review & editing:** Philippe Grandjean, Clara Amalie Gade Timmermann, Marie Kruse, Flemming Nielsen, Pernille Just Vinholt, Lasse Boding, Carsten Heilmann, Kåre Mølbak.

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
