## [Decision Letter · Decision Letter 0]

18 Nov 2020

PONE-D-20-32774

Severity of COVID-19 at elevated exposure to perfluorinated alkylates

PLOS ONE

Dear Dr. Grandjean,

Thank you for submitting your manuscript to PLOS ONE. After careful consideration, we feel that it has merit but does not fully meet PLOS ONE’s publication criteria as it currently stands. Therefore, we invite you to submit a revised version of the manuscript that addresses the points raised during the review process.

We look forward to receiving your revised manuscript.

Kind regards,

Jaymie Meliker, Ph.D.

Academic Editor

PLOS ONE

Journal Requirements:

2. In ethics statement in the manuscript and in the online submission form, please provide additional information about the patient records/samples used in your retrospective study. Specifically, please ensure that you have discussed whether all samples were fully anonymized before you accessed them and/or whether the IRB or ethics committee specifically waived the requirement for informed consent.

3. Please note that PLOS does not permit references to “data not shown.” Authors should provide the relevant data within the manuscript, the Supporting Information files, or in a public repository. If the data are not a core part of the research study being presented, we ask that authors remove any references to these data.

4.We note that you have indicated that data from this study are available upon request. PLOS only allows data to be available upon request if there are legal or ethical restrictions on sharing data publicly. For more information on unacceptable data access restrictions, please see http://journals.plos.org/plosone/s/data-availability#loc-unacceptable-data-access-restrictions.

5.Thank you for stating the following in the Competing Interests section:

[Apart from PG having served as health expert in lawsuits on environmental contamination, which does not affect the adherence to all PLOS ONE policies, the authors have no competing interests to declare, financial or otherwise. ].

Reviewers' comments:

Reviewer's Responses to Questions

**Comments to the Author**

1. Is the manuscript technically sound, and do the data support the conclusions?

Reviewer #1: No

Reviewer #2: Yes

2. Has the statistical analysis been performed appropriately and rigorously? 

Reviewer #1: Yes

Reviewer #2: Yes

3. Have the authors made all data underlying the findings in their manuscript fully available?

Reviewer #1: No

Reviewer #2: No

4. Is the manuscript presented in an intelligible fashion and written in standard English?

Reviewer #1: Yes

Reviewer #2: Yes

5. Review Comments to the Author

Reviewer #1: The authors report associations between PFAS and clinical progression of COVID19 among 323 Danish patients with SARS-COV-2 infection and archived blood specimens. The report focuses on higher odds for more severe clinical COVID19 disease associated with greater concentrations of blood PFBA, a short-chain and short-lived PFAS, in covariate adjusted models. Yet, consistent results are also presented for a “protective” PFHxS association, stronger it appears than that for PFBA. Despite the highly selected nature of the study population (as acknowledged by the authors), and the limited sample size, this paper has the potential to make an important contribution to the developing PFAS-human immunotoxicity literature, and especially its impact on the COVID19 pandemic. Still, there are several points that will benefit from additional development and other that merit clarification. Overall, a more balanced interpretation of the literature and the current results will be helpful.

Major points:

1. There are no line numbers or page numbers, this makes review/providing feedback challenging

2. Although the terms “ordered” and “ordinal” logistic regression are used throughout, I believe these to be synonymous. If so, the interpretation of the odds ratio from the ordinal logistic is generally for the cumulative odds; i.e., falling into a higher ordered category or less relative to the reference group. This is a subtle but important distinction of interpretation that should be clarified throughout.

3. I’m not sure that the title is appropriate; the outcome does not appear to be “severity of COVID-19” per se (e.g., viral load, etc.), given that comorbidities may also play an important role in the extent of treatment. The author might consider revising the title to more closely reflect the study outcome, such as “clinical course of COVID-19 infection” or something along those lines.

4. The links to air pollution studies, seem tenuous at best. The paper starts out with a focus on air pollution and limitations to the exposure assessment strategies that have been implemented for studies of COVID19. This seems like an “apples to oranges” comparison, and leaves the reader anticipating a study of air pollutants, not PFAS. I do appreciate the use of air pollution studies as a “proof of concept” but at present the focus is too great in my opinion.

5. Methods, “…we also included subjects, mainly those not hospitalized, whose plasma…” How many? While agreed that this may be less of an issue for the long-chain PFAS with ½-lived measured in years, even this difference might introduce or obfuscate a signal if systematically different between those with more/less severe disease (essentially a selection bias). Furthermore, the serum half-life for PFBA is measured in days. Is this a problem in terms of potential iatrogenic exposures to PFAS compounds during hospital stay/treatment among those with more severe disease? Specifically, where is PFBA used/found?

6. Methods, Chemical Analysis, “…were replaced by LOD/2…” Imputation of values below the LOD can lead to bias, in particular with a large proportion of values imputed (this shifts the mean and truncates the variance). The authors should repeat the analysis using the original, machine-read values for the analysis.

7. Methods, Statistical Analysis, “Associations between comorbidities…” Why can’t the results be presented without the corresponding cross-tabs, as differences with confidence intervals for example?

8. Methods, Statistical Analysis, A brief description of ordered (ordinal?) logistic regression could be provided as this approach is uncommon. While two terms are used here and in the abstract I am under the impression that they are synonymous, though unclear (see comment #2 above). Please clarify.

9. Methods, Statistical Analysis, “The assumption of PFAS linearity…” With only n=323, a more liberal p<0.10 seems more appropriate to assess PFAS linearity if a potential non-linear association is suspected. While I appreciate the latter when it comes to endocrine disrupting effects I am not familiar with non-linear associations reported for immune effects; is this exploratory? Still, this would seem to have been infeasible for dichotomized PFBA and PFBS, a limitation of the approach.

10. Methods, Statistical Analysis, “Potential confounding variable were identified based on a priori…” What a priori knowledge, specifically? What criteria to define a confounder and what supporting literature/citations?

11. Methods, Statistical Analysis, Did the authors adjust for plasma/serum albumin or renal clearance? What is the relationship between serum albumin and kidney function and COVI19 or and COVID19. Reports of compromised renal function among COVID19 patients might impact PFBA clearance for example (in particular given the short half-life), leading to reverse causal association. While the latter is less likely for inflated albumin due to infection (as the other PFAS were not associated with more severe disease progression), the issue merits attention.

12. Method, Statistical Analysis, How were diagnoses confirmed? Is there a difference in the diagnosis among those hospitalized vs. those not hospitalized?

13. Methods, Statistical Analysis. Was adjustment made for socioeconomic status? This is an important predictor of COVID19 outcomes in some countries and often a predictor of environmental exposures, including PFAS. If not adjusted, the authors should at least touch upon the relevance/importance.

14. Discussion, “Among…the PFASs, presence of detectable…” PFHxS showed the strongest association according to Table 4.

15. Discussion “The associations of PFASs…” The results do not support this conclusion, only PFBA above the LOD was associated with a more severe clinical course of COVID19, the other PFAS, including the persistent long-chain PFAS, were null or even protective (e.g., for PFHxS).

16. Discussion, this appears to be a (mostly) cross-sectional study; is reverse causation a concern here with respect in particular to renal clearance/serum albumin PFAS binding (see comment #11 above)

17. Discussion, “Low background exposure results…” How did the PFAS levels compare to other reports of associations with immune function?

18. Discussion, “Among immigrants, adverse associations appeared…” Is there any education/income data that might help to tease apart a sociodemographic disparity here? If not, is there national data that the authors can use to place this issue into a socioeconomic and/or residential context?

19. Conclusions, The authors should also report on the “protective” effect of PFHxS, a long-lived PFAS.

Minor points

1. Abstract: “Methods”: “…at the background exposures…” Unclear.

2. Abstract, “Results”: increasing severities of disease; again, does the outcome variable truly capture “disease severity.”?

20. Introduction, I am somewhat concerned about reliance on citing preprints (four of these that I found) given that these publications have not yet been “vetted” by peers and very well may change, in some circumstances dramatically, before eventual publication. This is analogous to citing four “submitted” papers, appropriate on occasion and perhaps this is one of those occasions. Given the novelty of the COVID19 pandemic and the nascent literature I’m not quite sure how to get around this but perhaps the papers can be cited in text as “submitted” or “preprint” to clarify for the reader.

3. Introduction, “…which accumulates in the lung..” This was shown by a single study in a single Spanish population. While I agree that it is important a very important point, it has not been conclusively established to my knowledge and should be reworded to reflect this, such as “..shown to accumulate in the lung.”

4. Introduction, a more comprehensive treatment of the PFAS immune function literature should be undertaken. Specifically, is the specific immune response to COVID19 threatened by PFAS according to previous work or is this a pending data gap too?

5. Methods, is exogenous contamination of specimens, for example Teflon caps, a concern for blood specimens?

6. Methods, “…no hospital admission and completed infection within 14 days of testing positive…” How was this determined?

7. Methods, is there a sufficient number of participants to stratify according to presence absence of obesity/chronic disease? Even in unadjusted analysis? Some prior work indicates difference in male/female PFAS-immunotoxicity; any difference between PFAS associations in men/women in this study?

8. Methods, Chemical Analysis, “Succeeding series…” Unclear.

9. Methods, Chemical Analysis, “…for PFAS concentrations, including…” Why these five PFAS specifically?

10. Methods, Chemical Analysis, “Because more than half the PFBA and PFBS…” PFBS was not mentioned in the “Chemical Analysis” section above or reported in the results as far as I can tell.

11. Methods, Statistical Analysis, “…presence of comorbidities…” What comorbidities did participants have, specifically?

12. Methods, Statistical Analysis, “…demographic groups…” What demographic groups?

13. Methods, Statistical Analysis, “The test could not be fitted…” Why not?

14. Methods, Statistical Analysis, “Among those of Western European national origin…” This seems better suited to the Results section.

15. Statistical Analysis, “…risk of certain chronic disease…” Which chronic diseases, specifically?

16. Results, “Correlated well..” Unclear.

17. Discussion, The first paragraph should summarize the major findings/and implications of the research. This interpretation is better suited to later components.

18. Discussion, “… the only PFAS that is substantially accumulated in the …” This has been shown to substantially accumulate in one study, please clarify this point (see comment #3 above)

19. Discussion, “…major pathways that are predictive…” What pathways, specifically?

20. Discussion, “Thus older age and male sex…” These are also strong predictors of COVId19 severity, making the likely to be confounding variables and so adjustment was appropriate and the diminished associations simply reflecting unbiased effect estimates. This seems like circular reasoning and should be clarified or removed from the manuscript.

21. Discussion, “As PFAS exposure has been linked to important comorbidities…not be justified.” Agreed, suggest stratification/interaction analysis if plausible, but comorbidities are probably not confounders here. A DAG would help to clarify this issue.

Reviewer #2: Title: Severity of COVID-19 at elevated exposure to perfluorinated alkylates

The aim of the present study was to assess if elevated background exposures to immunotoxic PFASs are associated with the severity of COVID-19 development. PFAS levels were measured in plasma samples from 323 subjects aged 30-70 years with known SARS-CoV-2 infections. Logistic regression analyses were performed to study the association between PFAS levels and disease severity. Elevated plasma-PFBA concentrations were associated with an increased risk of more severe course of CIVID-19. Since there are studies showing that PFASs can cause immunosuppression in humans, the present study may contribute with valuable information on the interplay between environmental chemicals – immune responses – disease severity of Covid-19.

Abstract

A1. “The course of coronavirus disease 2019 ( COVID-19) seems to be aggravated by air pollution, and some industrial chemicals, such as the perfluorinated alkylate substances (PFASs), are immunotoxic and may contribute as well.” What is the last part of the sentence referring to (PFASs or some industrial chemicals)? The sentence should be rewritten to make it more clear.

A2. “We used ordinal and ordered logistic regression analyses to determine associations between PFAS concentrations and disease outcome.“ Maybe I am wrong here, but isn’t ordered logistic regression the same as ordinal logistic regression?

Materials and methods

M1. Statistical analysis, 2nd section: PFBS – write the full name. Alternatively delete since no findings on PFBS are shown in the manuscript.

Results

R1. “PFBA was lower, but the origin of the samples was only weakly associated with plasma-PFAS concentrations (Table 2).” Since PFBA is used as a binary variable (below/above LOD), the authors should consider showing PFBA as n(%) where n is the number of samples above LOD. Showing the levels as mean and 95% CI does not give any information that are helpful

when interpreting the findings in Table 2.

R2. “A more severe disease outcome was associated with higher plasma-PFBA concentrations, also after adjustment for all covariates (Table 4).” This statement should be moderated in that this association becomes non-significant after adjustment for covariates (1.62(0.99, 2.64)). Consider moving this sentence from the Discussion to the Results section “The associations of PFASs with a more serious course of COVID-19 are weakened after adjustment for covariates, and some regression coefficients and ORs are below 1.”

R3. “In additional analyses, marginal changes in the ORs occurred when plasma samples obtained more than 60 days before diagnosis were excluded.” This may not be enough to adjust for the short half-life of 72 hours for PFBA. See also comment D3 below.

R4. “If not adjusted for the presence of chronic disease, the adjusted OR for PFBA was 1.77 (95% CI, 1.09, 2.87).” What is the rationale for excluding chronic disease as a covariate in this analysis? Please give an explanation for not adjusting for chronic disease. See also comment D2 below.

R5. “In dichotomous analyses comparing severities of the disease, detectable PFBA in plasma also showed a clear association with a more severe clinical course of the disease, most pronounced for odds between hospitalization and admission to intensive care unit/death (data not shown).” Isn’t this the finding shown in Table 4 and already mentioned above? Please clarify.

R6. To make it easier for the readers to interpret all the findings reported in the Results section, the author should consider showing results of all additional analyses as supplementary material.

Discussion

D1. “Given the persistence of the PFASs in general, the unique retention of PFBA in lung tissue may offer a clue to interpreting the findings in this study.” This is a very interesting theory. Are there more information on this issue to support such a theory, e.g.:

- Are there any studies showing the correlation between blood and lung levels of PFBA? This would be interesting to know since you are looking at plasma levels and not lung levels of PFBA.

- According to Pérez et al. 2013, lung was the tissue showing the highest accumulation of PFASs. Of the PFASs included in the present paper, Pérez et al reported mean concentrations of PFBA and PFOS of 807 and 28.4 ng/g lung tissue. In addition, even though the percentage of samples with detected values of PFOA fell down to 45%, the contribution of PFOA to the total PFASs in lung was quite important, in comparison other tissues and analytes. Since PFBA was not the only PFAS to be retained in the lungs, the authors could include this information in their discussion.

D2. “The associations of PFASs with a more serious course of COVID-19 are weakened after adjustment for covariates, and some regression coefficients and ORs are below 1. However, adjustment for all covariates may result in over-adjustment bias. Thus, older age and male sex are known to be strong predictors of higher blood-PFAS concentrations, and simple adjustment for these factors could potentially result in a bias toward the null. As PFAS exposure has been linked to important comorbidities, such as diabetes and obesity, both of which may exacerbate the virus infection, adjustment for chronic disease may also not be justified. Leaving it out slightly strengthened the PFBA association with the disease severity.” The authors should consider using a more structured approach of selecting covariates to be included in the adjusted analyses and not just include/exclude covariates since such an approach may lead to both over- and under-adjustments. Using Directed Acyclic Graphs (DAGs) to select covariates to include in the adjusted analyses, may decrease the chance of adjustment bias. Entering the covariates in the present study into Dagitty.net, this simple DAG returns a minimal sufficient adjustment set for estimating the total effect of PFASs on disease severity that includes age and sex only. The authors should also include possible other important covariates they may have information on. The Dagitty.net code for the DAG is:

dag {

"Disease severity Covid-19" [outcome,pos="1.400,1.621"]

Age [pos="-2.025,-1.064"]

Diabetes [pos="0.150,-0.740"]

Obesity [pos="-0.917,-0.547"]

PFASs [exposure,pos="-2.200,1.597"]

Sex [pos="-1.758,0.282"]

Age -> "Disease severity Covid-19"

Age -> Diabetes

Age -> Obesity

Age -> PFASs

Diabetes -> "Disease severity Covid-19"

Obesity -> "Disease severity Covid-19"

Obesity -> Diabetes

PFASs -> "Disease severity Covid-19"

PFASs -> Diabetes

PFASs -> Obesity

Sex -> "Disease severity Covid-19"

Sex -> PFASs

}

D3. “In many cases, their plasma had been stored on previous occasions, and the PFAS concentrations may reflect slightly higher exposures in the recent past. Although adjustment for the time interval since sample collection was included in the analyses, its impact on the results was negligible.” Since PFBA has a half-life of only 72 hours, this may affect your findings in that in the “No hospitalization group”, the number of days between sampling and diagnosis was 355 (22.5, 639.5) whereas it was 0 for the two other categories of severity. PFHxS, PFOS, PFOA and PFNA have half-life ranging from 2.5 to 8.5 years and may be representative of the PFAS levels at time of disease, whereas PFBA levels resemble more a “snap-shot”. Thus, the PFBA levels up to 28 months before disease onset may not be a reliable measure compared to the other PFASs included in the manuscript. Excluding samples that were taken 60 days before diagnosis may not be sufficient to adjust for the short half-life of 72 hours for PFBA. The authors should expand their discussion accordingly.

D4. “Among immigrants, adverse associations appeared slightly stronger, also after adjustments, thus suggesting that national origin, perhaps as related to demographic or social factors, may result in a greater vulnerability to PFAS-associated aggravation of the infection. Difference in age, sex, or comorbidities did not explain this tendency, but is in agreement with previous findings of ethnic differences in vulnerability.” These findings are not included in the Results section. If possible, this data should be included in the manuscript or as supplementary data.

D5. In adjusted analysis, there is a statistically significant inverse relationship between PFHxS and disease severity (0.52 (0.29, 0.91)), as well as for persons of Western European origin (0.46 (0.23, 0.89)). This is not mentioned specifically either in Results or Discussion. The author should include text on these findings in both these sections.

Conclusion

C1. “Thus, given the immunotoxicity of the PFASs, exposure to these persistent industrial chemicals may contribute to the severity of COVID-19.” This statement should be moderated in that PFBA is statistically significant related to disease severity in the crude analysis only. Moreover, PFHxS is inversely related to disease severity. If the authors decide to keep this statement, a clarification should be added.

6. PLOS authors have the option to publish the peer review history of their article (what does this mean?). If published, this will include your full peer review and any attached files.

Reviewer #1: No

Reviewer #2: No

---

## [Author Response · Author response to Decision Letter 0]

12 Dec 2020

Responses to Reviewer Comments 

Reviewer #1: The authors report associations between PFAS and clinical progression of COVID19 among 323 Danish patients with SARS-COV-2 infection and archived blood specimens. The report focuses on higher odds for more severe clinical COVID19 disease associated with greater concentrations of blood PFBA, a short-chain and short-lived PFAS, in covariate adjusted models. Yet, consistent results are also presented for a “protective” PFHxS association, stronger it appears than that for PFBA. Despite the highly selected nature of the study population (as acknowledged by the authors), and the limited sample size, this paper has the potential to make an important contribution to the developing PFAS-human immunotoxicity literature, and especially its impact on the COVID19 pandemic. Still, there are several points that will benefit from additional development and other that merit clarification. Overall, a more balanced interpretation of the literature and the current results will be helpful.

Response: Thanks for the detailed and helpful comments. 

Major points:

1. There are no line numbers or page numbers, this makes review/providing feedback challenging

Response: We have added the requested format change.

2. Although the terms “ordered” and “ordinal” logistic regression are used throughout, I believe these to be synonymous. If so, the interpretation of the odds ratio from the ordinal logistic is generally for the cumulative odds; i.e., falling into a higher ordered category or less relative to the reference group. This is a subtle but important distinction of interpretation that should be clarified throughout.

Response: We have corrected ordinal to ordered logistic regression throughout the manuscript.

3. I’m not sure that the title is appropriate; the outcome does not appear to be “severity of COVID-19” per se (e.g., viral load, etc.), given that comorbidities may also play an important role in the extent of treatment. The author might consider revising the title to more closely reflect the study outcome, such as “clinical course of COVID-19 infection” or something along those lines.

Response: We appreciate the advice, but we have not studied in detail the clinical course over time. We believe that severity is a better choice for the title. 

4. The links to air pollution studies, seem tenuous at best. The paper starts out with a focus on air pollution and limitations to the exposure assessment strategies that have been implemented for studies of COVID19. This seems like an “apples to oranges” comparison, and leaves the reader anticipating a study of air pollutants, not PFAS. I do appreciate the use of air pollution studies as a “proof of concept” but at present the focus is too great in my opinion.

Response: We believe that the air pollution studies support the notion that community exposures to environmental contaminants can generally worsen the outcome of the COVID-19. Our own study adds to this association by documenting individual exposure levels, which could not be done in the air pollution studies. We have adjusted the wording so that it is clear that our work does not focus on air pollution. 

5. Methods, “…we also included subjects, mainly those not hospitalized, whose plasma…” How many? While agreed that this may be less of an issue for the long-chain PFAS with ½-lived measured in years, even this difference might introduce or obfuscate a signal if systematically different between those with more/less severe disease (essentially a selection bias). Furthermore, the serum half-life for PFBA is measured in days. Is this a problem in terms of potential iatrogenic exposures to PFAS compounds during hospital stay/treatment among those with more severe disease? Specifically, where is PFBA used/found?

Response: As indicated in Table 1, especially the non-hospitalized subjects had blood samples drawn much before the infection. We have now stratified the data and focused an analysis on samples collected close to the time of infection. The exclusion mainly of older blood samples results in stronger findings and also moves the OR for PFHxS closer to 1. Additional PFASs are shown in the Supplementary information.

6. Methods, Chemical Analysis, “…were replaced by LOD/2…” Imputation of values below the LOD can lead to bias, in particular with a large proportion of values imputed (this shifts the mean and truncates the variance). The authors should repeat the analysis using the original, machine-read values for the analysis.

Response: We appreciate the advice, but signals below the LOD are too uncertain, and there are therefore no machine-read values that can be used. This applies to PFBS as well in the Supplementary information.

7. Methods, Statistical Analysis, “Associations between comorbidities…” Why can’t the results be presented without the corresponding cross-tabs, as differences with confidence intervals for example?

Response: The comorbidities are now described in detail in the main text. Along with place of inclusion and COVID-19 severity, they are categorical variables, and thus cannot be shown as differences.

8. Methods, Statistical Analysis, A brief description of ordered (ordinal?) logistic regression could be provided as this approach is uncommon. While two terms are used here and in the abstract I am under the impression that they are synonymous, though unclear (see comment #2 above). Please clarify.

Response: We have included a brief description of ordered logistic regression in the methods section.

9. Methods, Statistical Analysis, “The assumption of PFAS linearity…” With only n=323, a more liberal p<0.10 seems more appropriate to assess PFAS linearity if a potential non-linear association is suspected. While I appreciate the latter when it comes to endocrine disrupting effects I am not familiar with non-linear associations reported for immune effects; is this exploratory? Still, this would seem to have been infeasible for dichotomized PFBA and PFBS, a limitation of the approach.

Response: We agree with the reviewer that a linear association is most likely, which is why we did not per default include loglinear PFAS or PFAS squared. However, as is standard when performing linear regression models, we tested our assumptions. With 323 individuals we believe that we would be able to detect a fairly strong non-linear associations using a significance level of 0.05. For dichotomized PFASs our interpretation was not based on a linear assumption, and there was thus no need to test the assumption. 

10. Methods, Statistical Analysis, “Potential confounding variable were identified based on a priori…” What a priori knowledge, specifically? What criteria to define a confounder and what supporting literature/citations?

Response: We have now emphasized in the text that the potential confounders were identified from the information and references already provided. 

11. Methods, Statistical Analysis, Did the authors adjust for plasma/serum albumin or renal clearance? What is the relationship between serum albumin and kidney function and COVI19 or and COVID19. Reports of compromised renal function among COVID19 patients might impact PFBA clearance for example (in particular given the short half-life), leading to reverse causal association. While the latter is less likely for inflated albumin due to infection (as the other PFAS were not associated with more severe disease progression), the issue merits attention.

Response: Given the conditions of the study of anonymized data, we did not have access to patient charts or individual data on kidney function markers. We have now treated kidney disease as a covariate separate from other chronic disease and commented on its association with plasma-PFAS concentrations. . 

12. Method, Statistical Analysis, How were diagnoses confirmed? Is there a difference in the diagnosis among those hospitalized vs. those not hospitalized?

Response: All diagnoses were obtained from the national register, which has a high validity, and there is no reason to suspect any difference in validity between groups of subjects included in the study. 

13. Methods, Statistical Analysis. Was adjustment made for socioeconomic status? This is an important predictor of COVID19 outcomes in some countries and often a predictor of environmental exposures, including PFAS. If not adjusted, the authors should at least touch upon the relevance/importance.

Response: We appreciate the comment, as it relates to an increased risk of infection associated with socioeconomic status. However, it is not clear from the literature whether it predicts a more serious disease as well. Due to the limitations of the register data available to us, we focused on national origin as the most important socioeconomic predictor of corona infection. In agreement with previous studies, we find lower PFAS exposures in subjects of other national origin, but also that this factor did not affect the disease severity. 

14. Discussion, “Among…the PFASs, presence of detectable…” PFHxS showed the strongest association according to Table 4.

Response: We have now clarified that we refer to positive associations, as reflected in the a priori hypothesis. As we are examining the possible association with immunotoxic exposures, the PFHxS association is not plausible. In the restricted analysis, we find PFHxS not to be a significant factor in regard to disease severity. The same applies to other PFASs in the Supplementary information.

15. Discussion “The associations of PFASs…” The results do not support this conclusion, only PFBA above the LOD was associated with a more severe clinical course of COVID19, the other PFAS, including the persistent long-chain PFAS, were null or even protective (e.g., for PFHxS).

Response: We agree that PFBA is in accordance with the hypothesis, but PFHxS is not (and is not accumulated in the lungs). The text has been adjusted for clarity, while taking into regard the further analyses that show no significant effect of PFHxS, nor other PFASs. 

16. Discussion, this appears to be a (mostly) cross-sectional study; is reverse causation a concern here with respect in particular to renal clearance/serum albumin PFAS binding (see comment #11 above)

Response: We refer to our response above, that the samples obtained early after diagnosis could not have been affected by pathologies occurring in late disease development. 

17. Discussion, “Low background exposure results…” How did the PFAS levels compare to other reports of associations with immune function?

Response: We hesitate to provide detailed information on exposures in other studies, as the data are already available to the reader. However, we have added a sentence to support the notion that the population is not highly exposed, while referring to national data from the U.S.

18. Discussion, “Among immigrants, adverse associations appeared…” Is there any education/income data that might help to tease apart a sociodemographic disparity here? If not, is there national data that the authors can use to place this issue into a socioeconomic and/or residential context?

Response: We refer to a previous study on ethnicity as a risk factor for the infection. Our own opportunities to explore these factors are limited due to the size of the material and to the circumstances of studying sensitive information anonymously. We note that ethnicity status was not an important predictor of the PFAS associations with disease severity. Further, as noted above, we did not have access to other socioeconomic data

19. Conclusions, The authors should also report on the “protective” effect of PFHxS, a long-lived PFAS.

Response: We already report this association, which was substantially weakened in the further analyses, and we note that a protective effect is counter to expectation.

Minor points

1. Abstract: “Methods”: “…at the background exposures…” Unclear.

Response: The term is commonly used to indicate non-occupational exposure not caused by local sources. We have commented on this in the Discussion and would prefer not to add additional wording here. 

2. Abstract, “Results”: increasing severities of disease; again, does the outcome variable truly capture “disease severity.”?

Response: We can’t think of a better word, as clinical course would seem misleading.

20. Introduction, I am somewhat concerned about reliance on citing preprints (four of these that I found) given that these publications have not yet been “vetted” by peers and very well may change, in some circumstances dramatically, before eventual publication. This is analogous to citing four “submitted” papers, appropriate on occasion and perhaps this is one of those occasions. Given the novelty of the COVID19 pandemic and the nascent literature I’m not quite sure how to get around this but perhaps the papers can be cited in text as “submitted” or “preprint” to clarify for the reader.

Response: We shall comply with the journal policy in this regard. We note that the COVID-19 pandemic has increased the need for access to relevant reports in advance of peer review that may substantially delay public access to the findings. Incidentally, two of the in-press references cited now have more definite journal information, and two of the initial three medRxiv articles are now available as journal articles, as indicated in the reference list, so there is now only one left.

3. Introduction, “…which accumulates in the lung..” This was shown by a single study in a single Spanish population. While I agree that it is important a very important point, it has not been conclusively established to my knowledge and should be reworded to reflect this, such as “..shown to accumulate in the lung.”

Response: The study referred to is the only one published so far and encompassed samples from 20 autopsies. PFBA had by far the highest concentrations in lung tissue, higher than any of the other 20 PFASs analyzed, and with a median much higher than other PFASs. We highlight that this is so far a single study. 

4. Introduction, a more comprehensive treatment of the PFAS immune function literature should be undertaken. Specifically, is the specific immune response to COVID19 threatened by PFAS according to previous work or is this a pending data gap too?

Response: We hesitate to expand on this information, as we are already referring to the NTP review. Also, the specific mode of action of PFAS immunotoxicity is unknown. 

5. Methods, is exogenous contamination of specimens, for example Teflon caps, a concern for blood specimens?

Response: All sample processing was done by robot, and Teflon caps are not used. 

6. Methods, “…no hospital admission and completed infection within 14 days of testing positive…” How was this determined?

Response: This information was retrieved from the patient register as described. 

7. Methods, is there a sufficient number of participants to stratify according to presence absence of obesity/chronic disease? Even in unadjusted analysis? Some prior work indicates difference in male/female PFAS-immunotoxicity; any difference between PFAS associations in men/women in this study?

Response: We believe that we already provide this information in regard to sex dependence, but we are restricted about tabular data on obesity. 

8. Methods, Chemical Analysis, “Succeeding series…” Unclear.

Response: We ran the analyses in sequence to avoid any risk of drift. 

9. Methods, Chemical Analysis, “…for PFAS concentrations, including…” Why these five PFAS specifically?

Response: These are the five that have been found to associate with immunotoxicity in humans, as we now clarify. 

10. Methods, Chemical Analysis, “Because more than half the PFBA and PFBS…” PFBS was not mentioned in the “Chemical Analysis” section above or reported in the results as far as I can tell.

Response: PFBS is left out in the main text, as it is not known to be immunotoxic in humans, see instead Supplementary information.

11. Methods, Statistical Analysis, “…presence of comorbidities…” What comorbidities did participants have, specifically?

Response: Given the conditions of the study, we were unable to extract information on specific comorbidities of individuals, and we therefore grouped the conditions, as described. 

12. Methods, Statistical Analysis, “…demographic groups…” What demographic groups?

Response: Thank you for pointing out this imprecision. It has now been clarified in the text.

13. Methods, Statistical Analysis, “The test could not be fitted…” Why not?

Response: This was most likely due to too many strata, but the omodel package in Stata does not provide detailed information why a fit was not achieved.

14. Methods, Statistical Analysis, “Among those of Western European national origin…” This seems better suited to the Results section.

Response: We appreciate the comment, but we believe that the descriptive information on the study population fits better in the Methods section. 

15. Statistical Analysis, “…risk of certain chronic disease…” Which chronic diseases, specifically?

Response: The text has now been expanded with the list of diseases. 

16. Results, “Correlated well..” Unclear.

Response: The explanation is given in the table that the sentence refers to.

17. Discussion, The first paragraph should summarize the major findings/and implications of the research. This interpretation is better suited to later components.

Response: We prefer to highlight the study purpose before summarizing the results. It is not clear why the reviewer prefers a different sequence of presentation. 

18. Discussion, “… the only PFAS that is substantially accumulated in the …” This has been shown to substantially accumulate in one study, please clarify this point (see comment #3 above)

Response: We believe that the present wording is appropriate and that the reference provided reflects the evidence. Please see our response to #3 above. 

19. Discussion, “…major pathways that are predictive…” What pathways, specifically?

Response: We hesitate to include a discussion of the pathways, as this matter is not central to our study, and relevant detail is provided in the reference cited. We now mention IL-17 as one of the pathways.

20. Discussion, “Thus older age and male sex…” These are also strong predictors of COVId19 severity, making the likely to be confounding variables and so adjustment was appropriate and the diminished associations simply reflecting unbiased effect estimates. This seems like circular reasoning and should be clarified or removed from the manuscript.

Response: This point is exactly what we want to make, i.e., that PFASs (except for PFBA) are generally higher in men and the elderly, and that fits the tendency observed for COVID-19 severity. 

21. Discussion, “As PFAS exposure has been linked to important comorbidities…not be justified.” Agreed, suggest stratification/interaction analysis if plausible, but comorbidities are probably not confounders here. A DAG would help to clarify this issue.

Response: We already tried out a DAG but found that it in this case is not providing clarity due to several factors of possible importance, and we therefore decided to leave it out.

Reviewer #2: Title: Severity of COVID-19 at elevated exposure to perfluorinated alkylates

The aim of the present study was to assess if elevated background exposures to immunotoxic PFASs are associated with the severity of COVID-19 development. PFAS levels were measured in plasma samples from 323 subjects aged 30-70 years with known SARS-CoV-2 infections. Logistic regression analyses were performed to study the association between PFAS levels and disease severity. Elevated plasma-PFBA concentrations were associated with an increased risk of more severe course of CIVID-19. Since there are studies showing that PFASs can cause immunosuppression in humans, the present study may contribute with valuable information on the interplay between environmental chemicals – immune responses – disease severity of Covid-19.

Abstract

A1. “The course of coronavirus disease 2019 ( COVID-19) seems to be aggravated by air pollution, and some industrial chemicals, such as the perfluorinated alkylate substances (PFASs), are immunotoxic and may contribute as well.” What is the last part of the sentence referring to (PFASs or some industrial chemicals)? The sentence should be rewritten to make it more clear.

Response: We have edited the text to highlight industrial chemicals.

A2. “We used ordinal and ordered logistic regression analyses to determine associations between PFAS concentrations and disease outcome.“ Maybe I am wrong here, but isn’t ordered logistic regression the same as ordinal logistic regression?

Response: Yes, as we have now clarified.

Materials and methods

M1. Statistical analysis, 2nd section: PFBS – write the full name. Alternatively delete since no findings on PFBS are shown in the manuscript.

Response: PFBS is included in the supplementary tables, as it is not known to be a human immunotoxicant, and it does not accumulate in the lungs like PFBA.

Results

R1. “PFBA was lower, but the origin of the samples was only weakly associated with plasma-PFAS concentrations (Table 2).” Since PFBA is used as a binary variable (below/above LOD), the authors should consider showing PFBA as n(%) where n is the number of samples above LOD. Showing the levels as mean and 95% CI does not give any information that are helpful when interpreting the findings in Table 2.

Response: We believe that the revised presentation satisfies this comment. 

R2. “A more severe disease outcome was associated with higher plasma-PFBA concentrations, also after adjustment for all covariates (Table 4).” This statement should be moderated in that this association becomes non-significant after adjustment for covariates (1.62(0.99, 2.64)). Consider moving this sentence from the Discussion to the Results section “The associations of PFASs with a more serious course of COVID-19 are weakened after adjustment for covariates, and some regression coefficients and ORs are below 1.”

Response: We agree that the two-sided p-value is marginally above 0.05 and have edited the text, also in accordance with the new and restricted analyses. 

R3. “In additional analyses, marginal changes in the ORs occurred when plasma samples obtained more than 60 days before diagnosis were excluded.” This may not be enough to adjust for the short half-life of 72 hours for PFBA. See also comment D3 below.

Response: We agree and have conducted further stratifications and now present further restricted analyses.

R4. “If not adjusted for the presence of chronic disease, the adjusted OR for PFBA was 1.77 (95% CI, 1.09, 2.87).” What is the rationale for excluding chronic disease as a covariate in this analysis? Please give an explanation for not adjusting for chronic disease. See also comment D2 below.

Response: Chronic disease was not a significant predictor (Table 2) and did not change the estimate, but the restricted analysis seems more appropriate.

R5. “In dichotomous analyses comparing severities of the disease, detectable PFBA in plasma also showed a clear association with a more severe clinical course of the disease, most pronounced for odds between hospitalization and admission to intensive care unit/death (data not shown).” Isn’t this the finding shown in Table 4 and already mentioned above? Please clarify.

Response: We have edited the text accordingly and now provide Supplementary Information.

R6. To make it easier for the readers to interpret all the findings reported in the Results section, the author should consider showing results of all additional analyses as supplementary material.

Response: We now include Supporting information. 

Discussion

D1. “Given the persistence of the PFASs in general, the unique retention of PFBA in lung tissue may offer a clue to interpreting the findings in this study.” This is a very interesting theory. Are there more information on this issue to support such a theory, e.g.:

- Are there any studies showing the correlation between blood and lung levels of PFBA? This would be interesting to know since you are looking at plasma levels and not lung levels of PFBA.

Response: The autopsy study did not include blood samples, unfortunately, so we cannot comment.

- According to Pérez et al. 2013, lung was the tissue showing the highest accumulation of PFASs. Of the PFASs included in the present paper, Pérez et al reported mean concentrations of PFBA and PFOS of 807 and 28.4 ng/g lung tissue. In addition, even though the percentage of samples with detected values of PFOA fell down to 45%, the contribution of PFOA to the total PFASs in lung was quite important, in comparison other tissues and analytes. Since PFBA was not the only PFAS to be retained in the lungs, the authors could include this information in their discussion.

Response: We appreciate the detail, but we are concerned about providing much additional information from a published study. PFBA by far reached the highest concentrations in lung tissue, and other PFASs were lower by more than an order of magnitude. 

D2. “The associations of PFASs with a more serious course of COVID-19 are weakened after adjustment for covariates, and some regression coefficients and ORs are below 1. However, adjustment for all covariates may result in over-adjustment bias. Thus, older age and male sex are known to be strong predictors of higher blood-PFAS concentrations, and simple adjustment for these factors could potentially result in a bias toward the null. As PFAS exposure has been linked to important comorbidities, such as diabetes and obesity, both of which may exacerbate the virus infection, adjustment for chronic disease may also not be justified. Leaving it out slightly strengthened the PFBA association with the disease severity.” The authors should consider using a more structured approach of selecting covariates to be included in the adjusted analyses and not just include/exclude covariates since such an approach may lead to both over- and under-adjustments. Using Directed Acyclic Graphs (DAGs) to select covariates to include in the adjusted analyses, may decrease the chance of adjustment bias. Entering the covariates in the present study into Dagitty.net, this simple DAG returns a minimal sufficient adjustment set for estimating the total effect of PFASs on disease severity that includes age and sex only. The authors should also include possible other important covariates they may have information on. The Dagitty.net code for the DAG is:

dag {"Disease severity Covid-19" [outcome,pos="1.400,1.621"]

Age [pos="-2.025,-1.064"]

Diabetes [pos="0.150,-0.740"]

Obesity [pos="-0.917,-0.547"]

PFASs [exposure,pos="-2.200,1.597"]

Sex [pos="-1.758,0.282"]

Age -> "Disease severity Covid-19"

Age -> Diabetes

Age -> Obesity

Age -> PFASs

Diabetes -> "Disease severity Covid-19"

Obesity -> "Disease severity Covid-19"

Obesity -> Diabetes

PFASs -> "Disease severity Covid-19"

PFASs -> Diabetes

PFASs -> Obesity

Sex -> "Disease severity Covid-19"

Sex -> PFASs}

Response: We are grateful for the advice and we did indeed use a DAG as part of the planning but decided against inserting it into the manuscript. 

D3. “In many cases, their plasma had been stored on previous occasions, and the PFAS concentrations may reflect slightly higher exposures in the recent past. Although adjustment for the time interval since sample collection was included in the analyses, its impact on the results was negligible.” Since PFBA has a half-life of only 72 hours, this may affect your findings in that in the “No hospitalization group”, the number of days between sampling and diagnosis was 355 (22.5, 639.5) whereas it was 0 for the two other categories of severity. PFHxS, PFOS, PFOA and PFNA have half-life ranging from 2.5 to 8.5 years and may be representative of the PFAS levels at time of disease, whereas PFBA levels resemble more a “snap-shot”. Thus, the PFBA levels up to 28 months before disease onset may not be a reliable measure compared to the other PFASs included in the manuscript. Excluding samples that were taken 60 days before diagnosis may not be sufficient to adjust for the short half-life of 72 hours for PFBA. The authors should expand their discussion accordingly.

Response: We agree on this point, and we have therefore included further restricted analyses that take into account the short elimination half-life of PFBA in blood. 

D4. “Among immigrants, adverse associations appeared slightly stronger, also after adjustments, thus suggesting that national origin, perhaps as related to demographic or social factors, may result in a greater vulnerability to PFAS-associated aggravation of the infection. Difference in age, sex, or comorbidities did not explain this tendency, but is in agreement with previous findings of ethnic differences in vulnerability.” These findings are not included in the Results section. If possible, this data should be included in the manuscript or as supplementary data.

Response: Thank you, we now include Supporting Information to justify this conclusion. 

D5. In adjusted analysis, there is a statistically significant inverse relationship between PFHxS and disease severity (0.52 (0.29, 0.91)), as well as for persons of Western European origin (0.46 (0.23, 0.89)). This is not mentioned specifically either in Results or Discussion. The author should include text on these findings in both these sections.

Response: As indicated above, we now emphasize that this association is counter to the a priori hypothesis, and it disappears in the restricted analysis, possibly because of higher concentrations in “old” samples. 

Conclusion

C1. “Thus, given the immunotoxicity of the PFASs, exposure to these persistent industrial chemicals may contribute to the severity of COVID-19.” This statement should be moderated in that PFBA is statistically significant related to disease severity in the crude analysis only. Moreover, PFHxS is inversely related to disease severity. If the authors decide to keep this statement, a clarification should be added.

Response: As responded above, we have modified the text.

---

## [Editor Report · Decision Letter 1]

17 Dec 2020

Severity of COVID-19 at elevated exposure to perfluorinated alkylates

PONE-D-20-32774R1

Dear Dr. Grandjean,

We’re pleased to inform you that your manuscript has been judged scientifically suitable for publication and will be formally accepted for publication once it meets all outstanding technical requirements.

Kind regards,

Jaymie Meliker, Ph.D.

Academic Editor

PLOS ONE
---

## [Editor Report · Acceptance letter]

21 Dec 2020

PONE-D-20-32774R1 

Severity of COVID-19 at elevated exposure to perfluorinated alkylates 

Dear Dr. Grandjean:

I'm pleased to inform you that your manuscript has been deemed suitable for publication in PLOS ONE. Congratulations! Your manuscript is now with our production department. 

Kind regards, 

on behalf of

Dr. Jaymie Meliker 

Academic Editor

PLOS ONE